# Sectoral Analysis of Landscape Interiors (SALI) as One of the Tools for Monitoring Changes in Green Infrastructure Systems

**Irena Niedźwiecka-Filipiak** *[ID], **Justyna Rubaszek**[ID], **Anna Podolska and Jowita Pyszczek**[ID]

Institute of Landscape Architecture, Wrocław University of Environmental and Life Sciences, Grunwaldzka 55, 50-357 Wrocław, Poland; justyna.rubaszek@upwr.edu.pl (J.R.); anna.podolska@upwr.edu.pl (A.P.); jowita.pyszczek@upwr.edu.pl (J.P.)
\* Correspondence: irena.niedzwiecka-filipiak@upwr.edu.pl

**Abstract:** The aim of this article was to present Sectoral Analysis of Landscape Interiors (SALI). This method uses the idea of a landscape interior understood as a fragment of a landscape perceived from the level of a person standing at a given point. The analyses were conducted in two stages: stage I—the quantitative stage, and stage II—the qualitative stage. The first part of the research was the analysis of the percentage share of particular elements of landscape interiors in the images perpetuated in photographs, taking into account their level of transparency. The second part was the assessment of their quality based on expert knowledge. The use of the SALI method in the context of the analysis of greenery changes over time was illustrated on the example of the landscape interior of the main street in the village of Psary in Poland. The research was carried out at a time interval of 10 years—for the years 2009 and 2019. The results of the study indicate very large changes and loss of greenery (especially trees) and the associated deterioration of the landscape. The findings confirm the suitability of the method in landscape research at a human scale.

**Keywords:** green infrastructure system; greenery changes; village; landscape interior; human eye-level view

## 1. Introduction

Landscapes are dynamic and subject to constant transformations—this characteristic is referred to by many researchers from various disciplines dealing with landscapes [1–7]. The variability of the landscape results from the continuous influence of natural and human factors, as indicated in the very definition of landscape adopted in the European Landscape Convention: "landscape is an area, as perceived by people, whose character is the result of the action and interaction of natural and/or human factors" [8]. Monitoring and analysis of the course of changes is an important research issue, as it is the basis for landscape protection, planning, and management. Landscape changes can be considered in the context of driving factors, processes, patterns, or their constituent elements. One of these important landscape elements is greenery. The study of changes in greenery takes on particular importance in the context of its role as a green infrastructure in adapting to climate change and minimizing its effects [9–15], as well as in the context of urbanization processes which very often result in the reduction of green spaces [16–19].

Two main trends can be distinguished in research on changes in greenery in the landscape. The first is based on analyses of land use and land cover. Advances in remote sensing (RS) and geographic information systems (GIS) has significantly contributed to the development of this trend [20–23] as well as to undertaking research work over large areas [24]. The second trend is the one where

research is carried out from the perspective of the human eye and photographs are used to record the landscape [25–27]. This research belongs to the "human-scale" research group, which Long and Ye defined as "a fine scale characterized by human body and its surroundings, i.e., a scale that can be directly visible, touchable, and appreciable in a person's daily life" [28].

Studies of greenery seen from a human perspective using photography are used to learn about human preferences and behaviours, e.g., with regards to sense of security [29], comfort and stress relief, well-being [30–35], or the frequency of use of a given space [36], aesthetic appreciation of landscape [37,38], or in combination with the aspect of the ratio of height to width of the street to assess the acoustic, visual, and audio-visual comfort levels [39]. Landscape analyses from a human perspective are also an important part of visual research in landscape architecture, e.g., as part of the assessment of new investments, protection of valuable views and landscapes particularly "vulnerable" to changes [40]. The least common research topic so far is quantitative studies of greenery perceived in a given view [41], and more specifically the analysis of these changes over time. This is due, on the one hand, to the complexity of the studies themselves, which Ye also pointed out, and, on the other hand, to the availability of materials from particular periods of time. The problem of recording changes in greenery from the perspective of a person looking at it will therefore be important both in particularly valuable landscapes, but also in so-called everyday landscapes, especially where there are high dynamics of change. Such landscapes include the landscapes of suburban areas where intensive development of buildings and infrastructure takes place, which in turn results in the reduction of open areas and their fragmentation [42–47], as well as contributes to the change of the landscape of built-up areas within the space of periurban villages. In this perspective, the context of the perception of this greenery by people who move along the streets in rural areas is also important. Particularly in suburban villages, the processes of densifying built-up zones and cutting down of deciduous trees bring the character of rural landscape closer to urban areas. For this reason, small forms of greenery such as pocket parks and roadside greenery are also gaining importance here [48].

The study on changes in the share of greenery in the landscape is also, and perhaps above all, important in the context of shaping green infrastructure systems [49–52]. Green infrastructure planning refers to both urbanized areas (city and local scale) and rural areas (regional scale) [53]. In scientific discussions, built-up areas of villages are often ignored. They can, however, form or constitute part of regional green infrastructure systems, as it is the case in the Green Infrastructure of Wrocław Functional Area (GI WFA) system [54].

In the adopted methods used in visual studies of the landscape seen from a human perspective, two basic approaches to data acquisition can be distinguished. The first one consists of taking photographs manually; the second one is collecting ready-made data from Google Street View (GSV). The first approach, despite the greater amount of time needed to go out into the field and take a photograph, allows for an accurate insight into the place and time of taking the photo. As a result, it is possible to compare photos taken in the same place over the years, at predetermined intervals. However, due to its time-consuming character, it is suitable for small-scale research projects, e.g., within a housing estate or village. The use of ready-made GSV images in the second approach significantly accelerates the research process and is therefore suitable for analyses of larger areas, as shown by Berland and Lange [55], using GSV for research in the area of three municipalities, or Ye et al. [41], who have conducted research on a Singapore-wide scale. Both panoramic images [56] as well as views with a smaller horizontal field of view are subject to analyses [57].

Landscape research using human-level images uses different framing methods [39]. The so-called "Green View" index (GVI) proposed by Yang et al. [57] and developed by Li et al. [39] is used to assess the amount of greenery in street landscapes. Li et al. [39] based their research on GSV panoramic images, which covered a range of 360° horizontally and 180° vertically. They cut out six photos with a range of 60° horizontally and each of them had additionally two vertical photos with a range of 45° above and 45° below the horizon (18 photos in total—the selected frames). This gave a wide viewing shot, which can be seen by a person walking along the street.

Another important issue in visual research is the very way of analyzing the view preserved in the photograph. Here, more "traditional" and "modern" approaches can be distinguished. The "traditional" approach is based on manual calculations—dividing a photograph into a geometrical grid and calculating in each of its cells the share of the examined element, as was the case in the studies of Shafer et al. [58], as well as North et al. [48]. The latter divided the photograph into 588 units. The areas in which the studied landscape element accounted for more than 50% were taken into account in the summary and calculation of their percentage share in relation to all grid cells. Various forms of greenery were specified as the studied elements: grass, lower ground vegetation, flowering plants, bushes, trees as well as hardscape, and water. Calculations of the share of landscape components in a given view can be facilitated by various computer applications, such as SegNet [59], FCN [60], Deep-Lab Larg-FOV [61], and DeconvNet [62]. Their application allows for a significant acceleration of calculations, e.g., by using SegNet [41], they were able to examine the whole of Singapore. Landscape research from a human perspective uses image processing software such as Adobe Photoshop. Using this program (using the magic wand tool and histogram function) it is possible to calculate the number of pixels contained in individual elements of a view, such as greenery, in relation to other elements [57,63]. Data obtained from the analysis of the flat image can be supplemented with airborne Lidar data. With airborne Lidar, it is possible to measure in detail individual elements including greenery, i.e., determine the shape, height, width of trees or shrubs, as shown by, e.g., Chen, Xu, and Gao [64].

In view of the above issues, the main aim of this article is to present the method of Sectoral Analysis of Landscape Interiors (SALI). This method was developed in the years 2000–2009 during the studies conducted in Poland under the Rural Renewal Program. At that time, the subject of analyses were the differences in the management of rural areas between historical and contemporary parts of villages, while at present this method has been used to assess changes in greenery over time [65,66]. The second aim of the study is to answer the question whether and to what extent the SALI method can become a tool to support the monitoring of greenery changes in urbanized interiors of villages located in green infrastructure systems.

## 2. Materials and Methods

The SALI method is used to assess the landscape from the human perspective (eye-level) and makes it possible to obtain data on the number and value of the elements that make up a given view and their mutual relationships and influence on the landscape. The method can be used to compare the landscape of different places at the same time period or to monitor changes in the same place at specific time intervals. The approach uses the concept of landscape interior understood as "the physiognomic surroundings of the place from which a person perceives the landscape" [67]. The term has been used for several decades in the Polish school of landscape research and refers to both urbanized and open landscapes to describe their physiognomy. It is used especially by scientists from the circles of architects and landscape architects [68–75].

The studies performed with the use of SALI method comprise:

- Quantitative analysis—this is the main part of the research, consisting of calculating the percentage share of individual components of the studied landscape interior;
- Qualitative analysis—performed as a complementary element of the first part of the study, including the assessment of the character of the walls of the studied landscape interior, the form and condition of its individual components, the relationships between them, their role in the landscape, as well as cultural values.

The research carried out using the SALI method is conducted in two stages: stage I—initial (preparatory stage) and stage II divided into two typically analytical stages (stage IIa and stage IIb), allowing for the assessment of the landscape interior and its individual constituents (Figure 1).

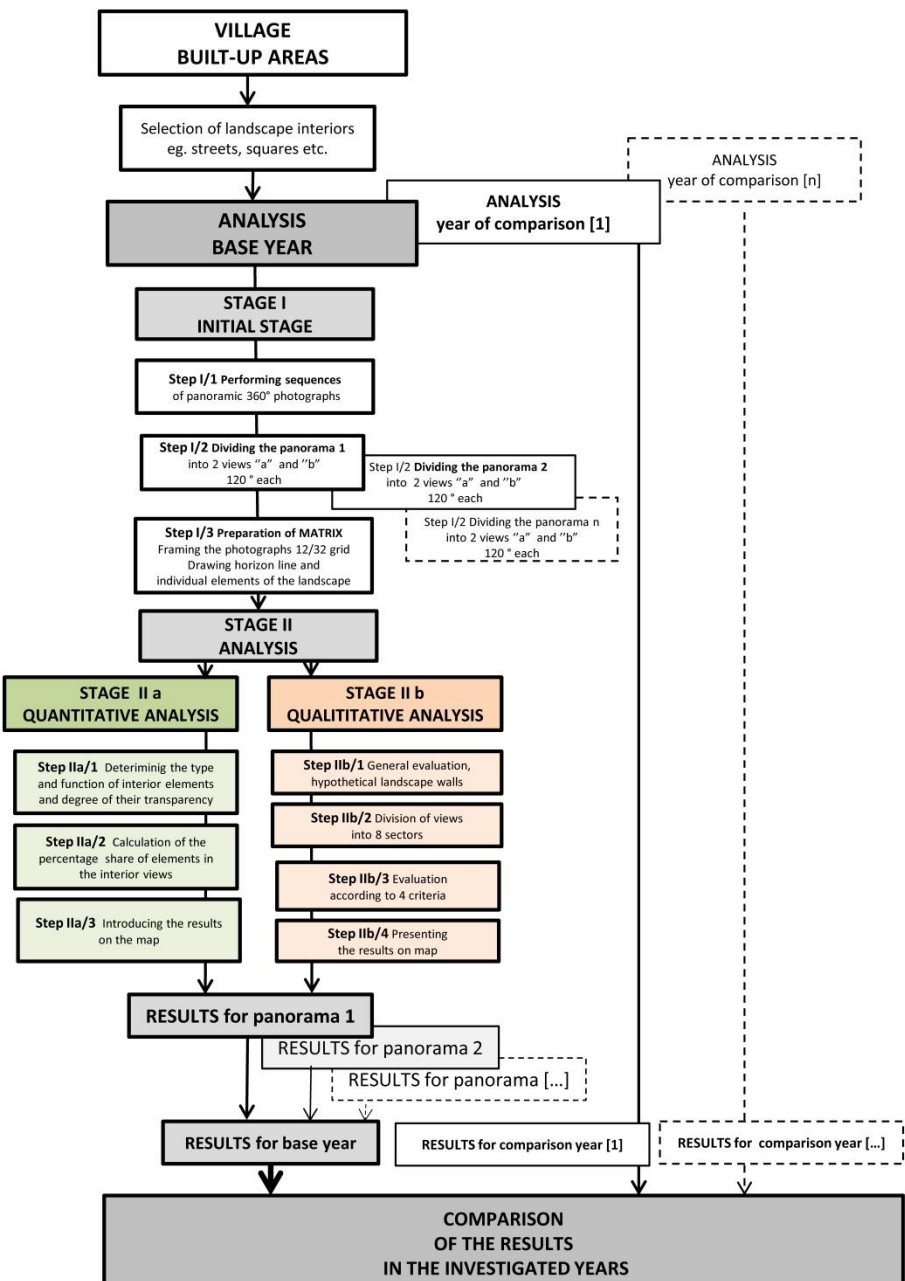

**Figure 1.** Diagram showing the procedure for performing tests using the Sectoral Analysis of Landscape Interiors (SALI) method.

## 2.1. Stage I: Initial Stage

The aim of the SALI investigations is to complement with 3D visual analyses the studies and landscape analyses carried out on two-dimensional maps. They consist of recording and evaluating landscape interiors seen by a person moving around a built-up area. Therefore, the research conducted so far has been carried out in the landscape interiors of streets. The analyses were carried out along them by taking a sequence of panoramic photographs (360°) at distances of 100–150 m (step I/1). The selection of the point where the photographs were taken was determined by the area conditions and the change in landscape physiognomy perceived by the observer. Photographs in the SALI method can be taken during the non-foliation and foliation period, which additionally makes it possible to

show the variability of the landscape over time, resulting from the passing of seasons. After taking a sequence of panoramic photographs, a matrix should be prepared for further research.

It is assumed that from each panoramic photograph taken, two fragments with a range of visibility of 120° should be extracted, representing a view in both directions into the street (step I/2). The analyses take into account fragments of the panoramic picture taken, which results, among others, from the observer's range of view. For most people, the area of good visibility, which makes it possible to distinguish between colors, is at an angle of 50–60°, and the size of the horizontal field of view in humans, which is called the binocular field of view, is 120° [76–78]. Assuming head and eye mobility, the range of visibility can reach 200°. The value of 120° (horizontal range of visibility of 120°) is considered optimal, but this does not mean that tests cannot be performed at a different angle.

After dividing the panoramas into two views, a matrix is to be prepared for further research (step I/3). In order to be able to compare both the quantity and quality of individual landscape elements, as well as their variability in time, on each of the photographs (a fragment of the panoramic picture) a line of the horizon is introduced, adopting a range of vertical visibility of 45°, and then a grid of frames with proportions: 12 units for height and 32 units for width, producing a division into 384 frames. The size and proportions of the grid result from earlier studies, when the basis for the analyses was one photo with an angle of 60° and the ratio of height to width 3:4, and this resulted in the present 12:16 ratio grid [79]. On the grid, a line between the fifth field of the grid counting from the bottom and the seventh field from the top is marked by bold or color change. The grid is applied by matching the marked line to the horizon line drawn earlier in the photo. When taking the position of the line on the grid, the golden ratio is used, as its use in landscape composition gives a sense of harmony and makes for its positive visual reception [80,81] The fascination with the golden ratio and its universality dates back to ancient times. Euclid defined it as: "A straight line is divided in a golden way when the ratio of the entire line to the larger segment is equal to the ratio of the larger to the smaller" [82]. Such a position of the horizon line also enables the visibility of most of the elements of the analyzed view. After superimposing the grid on the photo, it is corrected by cropping the fragments of the photo remaining outside the grid or complementing the missing ones. The width of the photo is fixed and invariable.

In photographs thus prepared, with the overlaid grid and the line of the horizon drawn, outlines of individual elements of the view are made. Due to the fragmentation of the structure of some landscape elements, when outlining the contours of the elements, some simplifications are made, e.g., entire crowns of trees are shown, not including details of particular branches or leaves; a similar principle is applied to shrubs and other forms of greenery. Electric power lines are also marked out based on the outermost wires, thus creating visible planes "occupied by technical infrastructure". Similarly, all fences are drawn as whole planes, without penetrating into their structure. A picture prepared in such a way becomes the matrix, which is the base for the next stage II (analysis), which is divided into two parallel parts IIa and IIb.

## 2.2. Stage IIa

The first analysis—quantitative—is to show how many and in what proportions particular elements of the landscape interior can be seen by a person moving along the street. On the prepared matrix, the cells obtained thanks to the outlines made are filled with color according to the following classification (step IIa/1):

1. Buildings, with division into: 1.1 residential, 1.2 utility, 1.3 facilities and services;
2. Greenery, with division into: 2.1 high, 2.2 medium height, 2.3 low;
3. Small architecture forms such as benches, garden houses and pavilions, fences;
4. Infrastructure, including visible elements of technical infrastructure;
5. Communication routes with division into: 5.1 roads with their movable and immovable elements, e.g., cars, 5.2 sidewalk pavements;

6.　　Water—watercourses and reservoirs;

7.　　Farmlands;

8.　　 Sky-canopy.

Quantitative calculations (step IIa/2) for individual elements in the examined view are performed using the traditional method of calculating their percentage share in individual grid cells, assuming that the grid cell size is 1 unit and its area is 1. What is significant here is that when calculating the percentage share of elements in particular grid units, using their generalized contours, the so-called transparency of those elements which partially cover those behind them is taken into account. Three degrees of transparency are adopted:

1.　　a high degree of transparency, e.g., leafless trees and deciduous shrubs, metal mesh fences, electric power lines, etc., allowing the elements behind them to be seen clearly (1st degree);

2.　　medium degree of transparency, e.g., deciduous or leafless trees and shrubs with leaves or leafless ones but with non-compact crowns, fences of different structures allowing the shapes and colors of the elements behind them to be seen but without any clear detail (2nd degree);

3.　　a low degree of transparency, e.g., elements which allow only the outlines of the elements behind them to be seen (3th degree).

What was also subject to consideration was how to treat moving elements of the landscape, such as vehicles, people and animals. For vehicles, a conventional degree of transparency was established: cars parked or passing through have first degree of transparency if they are parked or pass occasionally, second degree when individual cars are constantly present in the analyzed space, and third degree when it is impossible for a photograph to be taken due to at least one car being parked or passing through. The view area occupied by vehicles should be added to the group of elements no. 5 "road". People and animals are treated in a similar way and three degrees of transparency are also adopted here. Depending on the purpose of the study, the areas occupied by these objects can be counted in a separate group, e.g., as "moving animated landscape elements" or added to the already adopted groups. In this analysis, it was assumed that the view area occupied by people was added to the area in which they move, e.g., pavement, square, and animals, if any are found, to the area in which they were present.

Objects that completely cover what is behind them do not possess any degree of transparency, these are mainly buildings, evergreens, and other building structures of compact character. It is assumed that the degree of transparency is an estimation of the percentage of elements in the foreground that cover those in the background, so for the first degree it is 25%, for the second degree—50%, and for the third degree—75%. These values are estimated by experts performing the study or by a team under their supervision/guidance. The adopted values describing the degree of transparency of particular elements are important at the last most important stage of this part of the SALI method. This step IIa/2 consists of the calculation of the percentage share of individual elements in the entire view under analysis. The calculation consists of summing up the area of each of the 13 elements (grouped into 8 main groups) in the 384 individual units of the grid, taking into account their transparency:

$$X_i = \sum_{n=1}^{384} (\%a_n + \%b_n)\text{—total area of element examination,}$$

where: $i$ is the constituent element of the street interior, $n$ is the grid unit number, $a$ is the surface area of the grid unit under analysis that is completely covered by a given element, $\%b$ is the sum of transparent surfaces in which a given element is visible, $X$ is the total area of the item in the view under examination.

In step IIa/3, the functions of individual elements as defined in step IIa/1 can be displayed on the map.

### 2.3. Stage IIb

The second analysis—qualitative—is complementary to the analyses of stage I and aims to determine the character of the landscape interior of the selected street, the relationships between its elements, and the assessment of its value according to the adopted criteria. The basis for the research, as in stage I, is the matrix prepared in the initial stage.

At the beginning, the experts make a general subjective evaluation of each element of the interior under assessment with regard to its form, its embedding in the surroundings, and its technical condition, using three evaluation criteria: good, neutral, bad (step IIb/1). Then, a pattern of hypothetical landscape walls of the street interior should be drawn on the matrix. Bogdanowski's [67] approach is used to define and interpret the walls of the landscape interior, which defines them as definite, non-definite objective, and non-definite subjective. The author adopts the criterion of free spaces between the elements that make up the walls. Up to 30% of the clearances are definite walls, between 30% and 60% of the clearances are non-definite objective walls, whereas more than 60% of clearances indicates a wall referred to as a non-definite subjective wall. In the case of analyses of landscape interiors of village streets, the type of hypothetical walls depends on the region and nature of the built-up area of a given village and its specific layout, e.g., whether the built-up area is compact, loose, or dispersed. The number and layout of trees and shrubs along streets are also important.

Once the hypothetical landscape walls are drawn on the matrix, it is divided into eight sectors (step IIb/2), which correspond to a 15° visibility range, and another evaluation is performed according to four criteria: landscape values, form of the constituent elements, cultural value, and technical condition (step IIb/3). The assessment is carried out in two adjacent sectors together, i.e., within a range of visibility of 30°. For a more objective result to be achieved, the assessed visibility scopes overlap the sectors (they share 15° each). Evaluation criteria (each on a four-level scale) are presented in Table 1.

**Table 1.** Criteria of evaluation in individual sectors (step IIb/3).

| Criteria | State/Condition | Number of Points | | | |
|---|---|---|---|---|---|
| | | 0 | 1 | 2 | 3 |
| Landscape values—presence of dominants, accents, landscape gates, view openings, etc. | presence of more than two elements | | | | X |
| | presence of two elements | | | X | |
| | presence of one element | | X | | |
| | None | X | | | |
| Form of constituent elements—the assessment focused on the shape of all components of the interior, i.e., buildings and greenery | all building structures have appropriate forms and proportions, and the form of trees and shrubs corresponds to their correct/natural proportions | | | | X |
| | a single interior element differs in form or proportion from other elements belonging to the same category | | | X | |
| | several elements of the interior have a form or proportion that differs in character from other elements of the same category | | X | | |
| | many interior components are of a form or proportion different from other components of the same category | X | | | |
| Cultural value—viewed as a reference to the tradition of the place buildings and greenery | conspicuous, eligible character of the village | | | | X |
| | single elements differ from the rural character | | | X | |
| | single elements point to a rural character | | X | | |
| | lack of legibility of the rural character | X | | | |
| Technical condition—the assessment covered all components of the interior, including buildings as well as greenery | all elements are in good technical condition | | | | X |
| | a single element is in poor technical condition | | | X | |
| | several elements are in bad technical condition | | X | | |
| | many elements are in bad technical condition | X | | | |

The above evaluation should be graphically presented in the form of step diagrams displayed under the matrix. The result of each assessment is shown graphically in the form of a bar/dash in the adopted colors, a different color for each of the five criteria. The width of the bar/dash corresponds to

the points assigned (from 0 to 3). In addition to the ratings shown as bars/dashes, the total final grade for each sector must be entered. The final score is presented in the next diagram below the step diagram, which shows the average score for the sector from the two grades obtained by overlaying/overlapping. The grade results obtained in this way transferred to the map show the sectors, determined by the visibility ranges that need to be protected (high grade) or subject to taking measures due to the low grade achieved (step IIb/4).

The presented SALI method can be used both for studies of the current state of the landscape and for comparative studies carried out at specific time intervals, the results of which make it possible to present the variability of the landscape over time. However, comparative research is possible provided that historical photographs are available and/or planned for the future.

## 3. Results

The SALI method was used to investigate changes in the landscape of the village of Psary (Poland). This village is located within the urban fringe of Wroclaw and is directly adjacent to the city borders. This area is part of the planned first ring of the Green Infrastructure System of the Wroclaw Functional Area (GI WFA). The project of the GI WFA was created in 2014 as the basis for protecting and shaping greenery in the functional area of the city [83]. The system comprises both built-up and open areas, with different forms of natural and semi-natural land cover, divided into landscape functional units (LaFU). The LaFU unit in which Psary is located is threatened with degradation by the progressive development of buildings and infrastructure and requires protective and strengthening measures to be taken, among others, by preservation of open areas, introduction of greenery elements along with newly emerging investments, supplementing the already developed areas with greenery [54] (Figure 2).

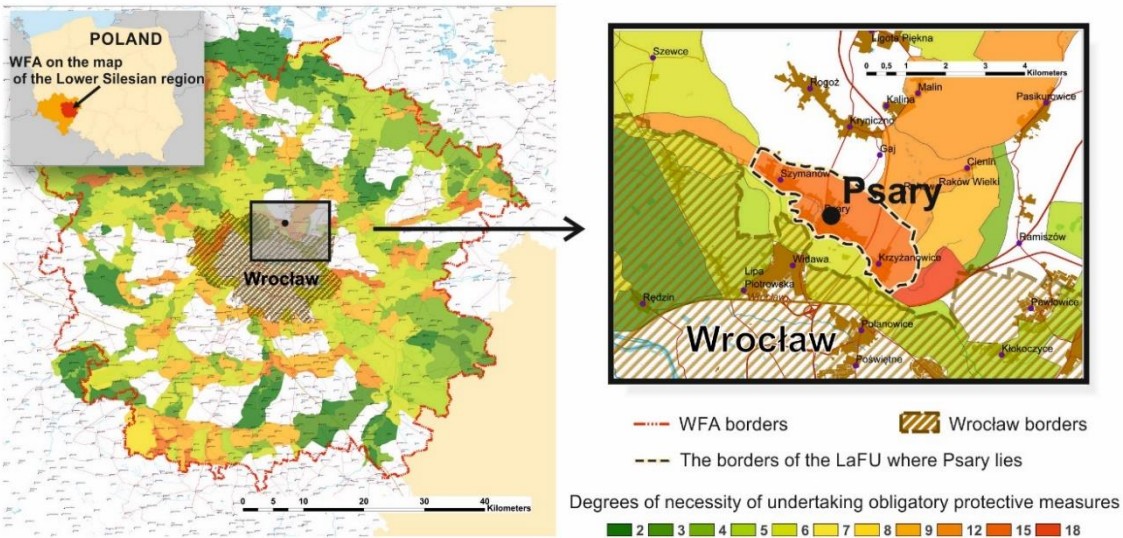

**Figure 2.** Location of the village of Psary in the urban fringe of the city of Wrocław (Poland), in the Green Infrastructure System of the Wroclaw Functional Area (GI WFA) and its unit, which is at risk of degradation [54].

The new commercial buildings, which have been developing intensively since 2000, have a particularly huge impact on the village landscape. From the side of Wrocław (south), a large-size shopping center and a car showroom have been built, while from the north, along the exit road towards Poznań, large warehouses and reloading halls are gradually being built. Unfortunately, the village does not have a comprehensive local spatial plan, which is a uniform document that could define its future sustainable development in an obligatory and a coherent way under the Polish law.

Appropriate management of built-up areas in villages, taking into account greenery, especially trees, is important for strengthening the GI WFA system. However, as shown by the research carried out in the public areas of the village of Psary—in Główna street in the years 2003–2012—there was a significant reduction in deciduous trees, from 108 to 82, while the number of coniferous trees remained unchanged at a constant level of 27. These studies were carried out on the basis of map analysis and field inventories. The reason for the changes in greenery at that time was the road lane modernization works [84]. The SALI method was used for the same street to examine what changes occurred in greenery and other landscape components in the next time period (2009–2019). This time it was also important to document and analyze the changes from the perspective of a moving person-.

Główna Street, as mentioned above, performs the function of the main public space of the village. Both residential and commercial buildings are located there, including a large shopping center. It is also one of the main exit routes from the city to the north, which is used for commuting from the suburbs. Every day it is used by both the inhabitants of Psary and people passing through the village, being a place of more or less conscious observation of the landscape by a significant number of people.

The studies were carried out in autumn 2009 and 2019. Panoramic pictures (360°) were taken in the same seven locations. The distances between them ranged from 100 to 150 m, which was due to the terrain conditions and change in the physiognomy of the landscape (Figure 3).

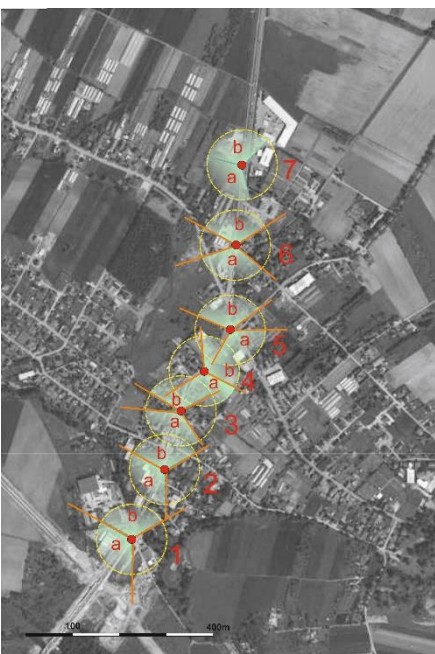

**Figure 3.** Location of seven sites along the main street of the village of Psary. Symbols from 1 to 7 indicate the points at which the picture were taken; symbols a, b indicate the fragments of the panoramic picture selected for analysis (120° views).

Detailed results of the analyses will be shown on the example of one selected location, no. 3. For each of the six remaining locations the tests were analogous. Additionally, for this single site no. 3, a survey was carried out in the summer of 2019 during the period of full foliage, due to clear differences in visibility of individual landscape elements between the foliage and the non-foliage period.

Figure 4 shows the panoramic photos (360°) prepared at stage I of SALI, taken at point no. 3, and the views "a" and "b" framed within them.

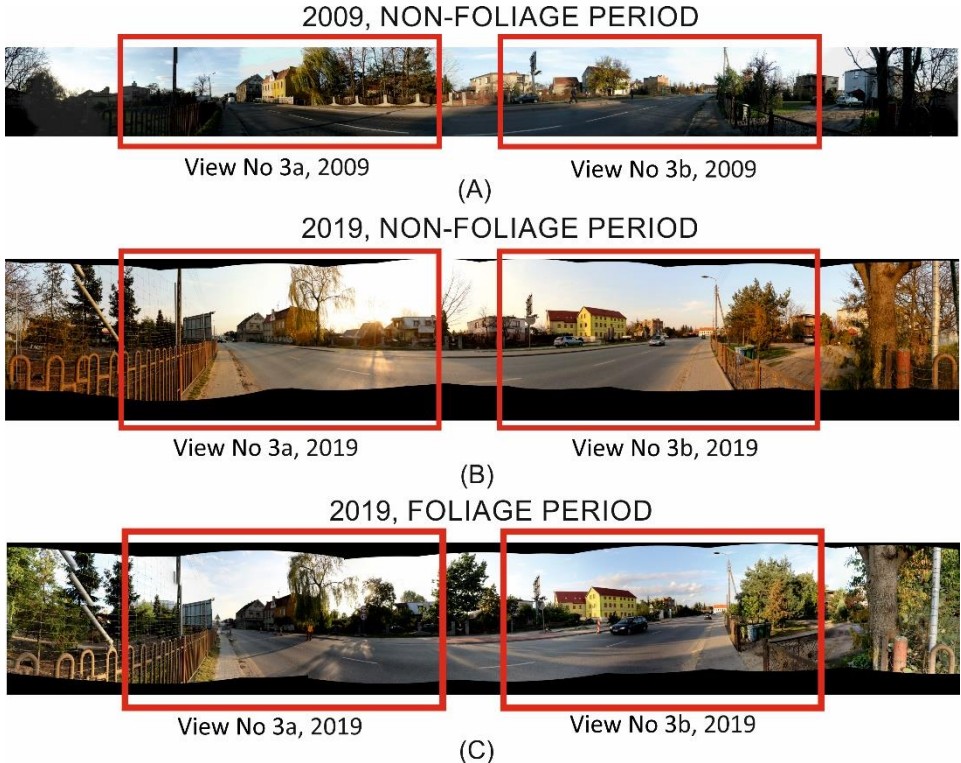

**Figure 4.** Stage I of SALI. Panoramic photographs of the interior of Główna Street in the village of Psary at point no. 3, showing views "a" and "b" with a range of 120° for detailed analysis (red box) (**A**) in the non-foliage period in autumn 2009. (**B**) in the non-foliage period in autumn 2019. (**C**) during foliage period of the summer 2019.

On the basis of such framed pictures, matrices were prepared and a quantitative analysis (stage IIa of SALI) was carried out, taking into account the degree of transparency of individual elements in a given view (Figures 5–7).

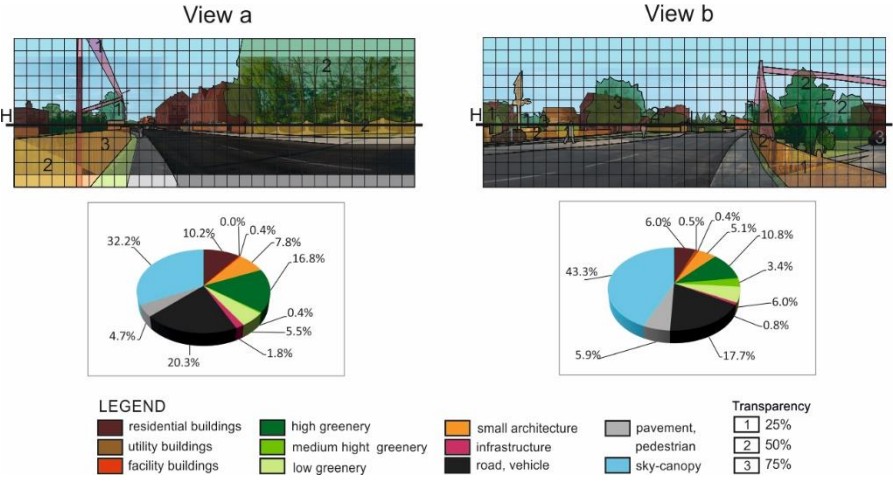

**Figure 5.** Stage IIa SALI quantitative analysis in interior 3—views "a" and "b"; autumn 2009 (non-foliage period).

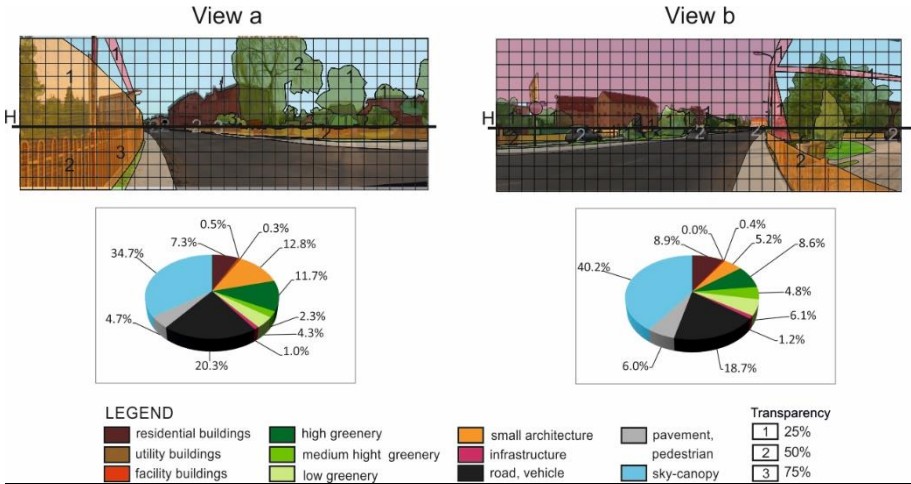

**Figure 6.** Stage IIa SALI quantitative analysis in interior 3—views "a" and "b"; autumn 2019 (non-foliage period).

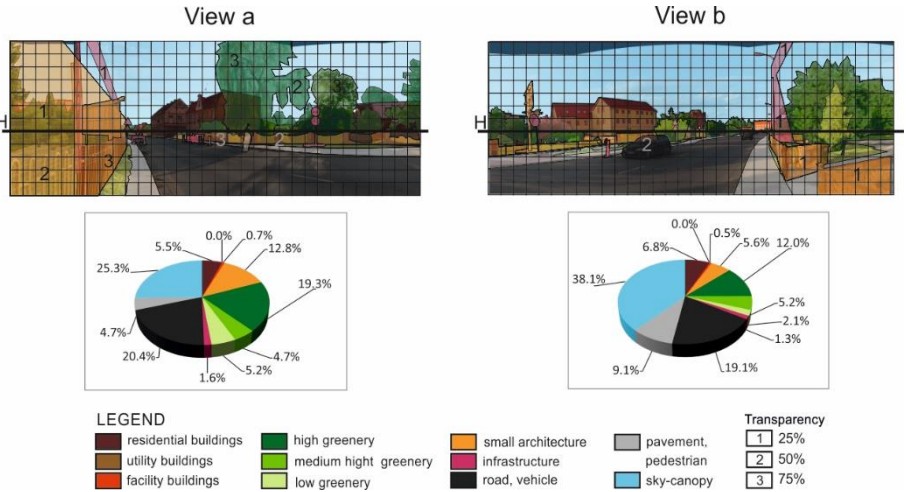

**Figure 7.** Stage IIa SALI quantitative analysis in interior 3—views "a" and "b"; summer 2019 (foliage period).

In the non-foliage period, the deciduous greenery received, in most cases, first and second degree of transparency. Visible fences sometimes made it possible to see what was behind them, and so they were given first, second, or third degrees of transparency. During the foliage period, the transparency of deciduous greenery changed mostly to second and third degrees or became totally absent when the trees or shrubs completely covered the view of what was behind them. Then, the results of the percentage of the view area occupied by high greenery in both years during the non-foliage periods were compared. It was found that this percentage had decreased over the 10-year period in both views "a" and "b". In the first view, there was a decrease of about one third, from 16.8% to 11.7%, and in view "b" by approximately one quarter, from 10.8% to 8.6%. There was, on the other hand, an increase in the amount of medium height greenery (shrubs), from 0.4 to 2.3 and from 3.4 to 4.8, and a slight decrease in the percentage of low greenery in view "a" from 5.5 to 4.3, and in view "b" it remained stable at approximately 6% in the autumn of 2019, just as in the autumn of 2009.

A comparison of the results of tests for point no. 3 performed in the summer of 2019 and the autumn of the same year indicates a clear difference in the percentage of high greenery in the foliage period compared to the non-foliage one. The percentage share was higher by about 60% in view "a" and about 50% in view "b" and it amounted to 19.42% and 12.02%, respectively. These differences were mainly due to the change in the percentage share of buildings and the sky in both views—the buildings

and the sky during the summer were covered with greenery. This study also indicates the necessity to carry out comparative studies using the SALI method in the same period of the year. The results of the percentage share of components in the area of all seven views in 2009 and 2019 are shown in Figure 8, in the form of pie charts for individual years. In all the 14 analyzed views located along Główna Street, two landscape elements included in the SALI method were missing, i.e., elements from group no. 6—water and from group no. 7—farmlands.

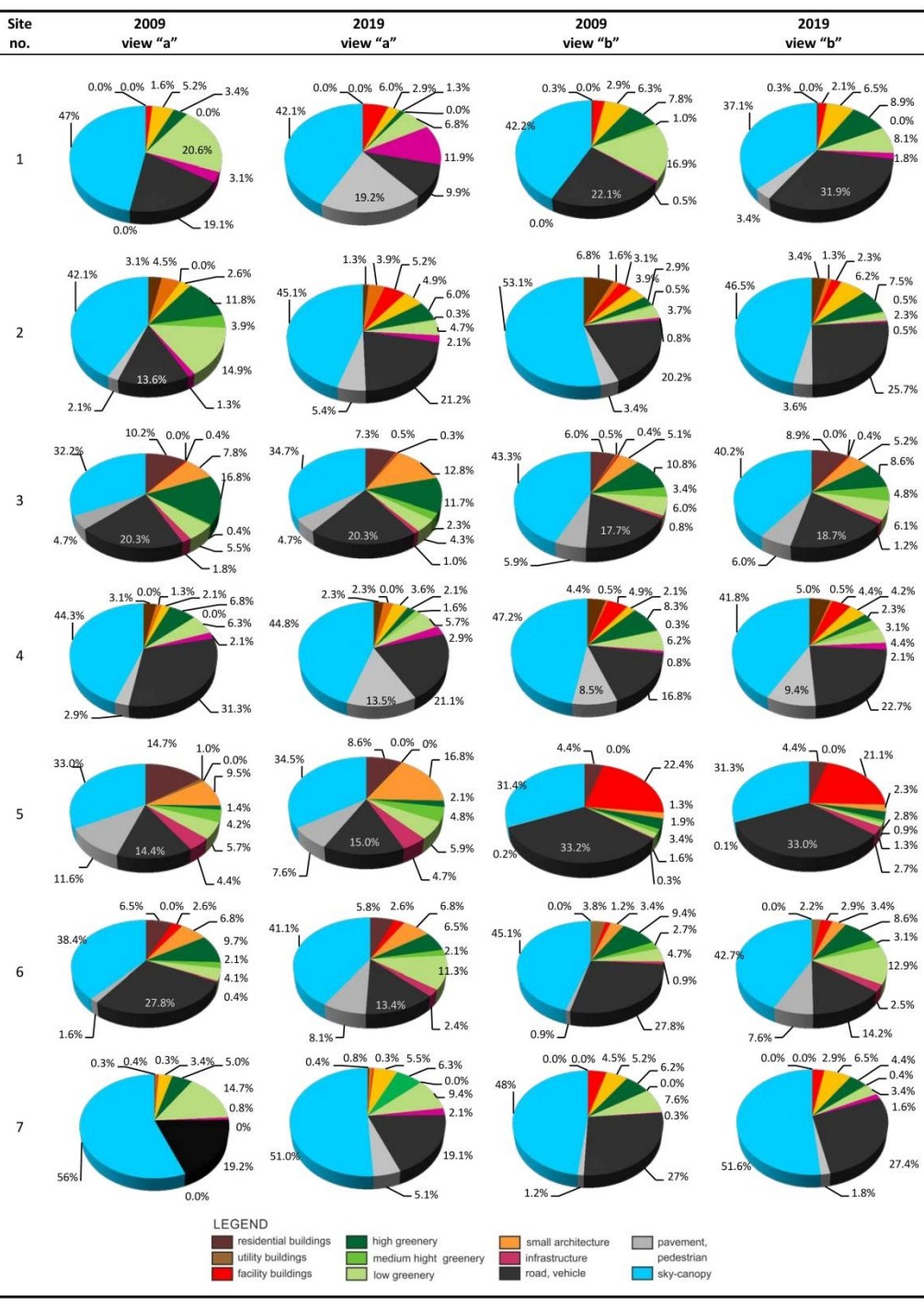

**Figure 8.** Diagrams showing the percentage share of individual components in views "a"and "b" at the 7 points.

On the basis of the obtained results (stage IIa), comparisons of greenery changes were made with the division into high greenery (trees), medium height greenery (shrubs), and low greenery (Figure 9). This aimed to compare the changes in the percentage of greenery in the seven analyzed points of Główna Street, taking into account the variability of views, which were observed by a walking person. List "a" presents a sequence of views from the northern entry to Psary in the direction of the southern exit from Psary to Wrocław (points 7 to 1), and list "b" is a sequence of views observed by a walking person moving along Główna Street in the opposite direction (points 1 to 7).

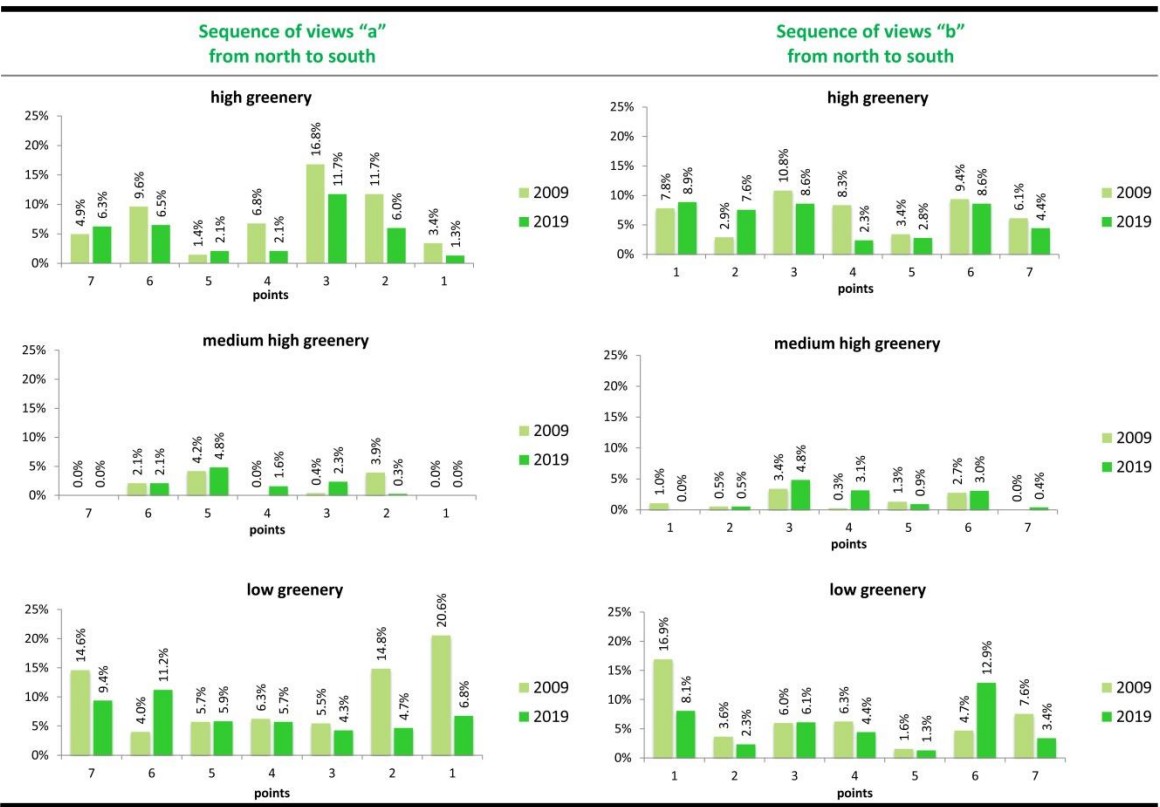

**Figure 9.** Percentage share of high (trees), medium (shrubs), and low greenery in the view sequences along Główna Street in Psary.

The results presented in the diagrams show that the visibility of trees in the seven analyzed points in 2009 was higher in a significant part of both view sequences "a" and "b" than in 2019. The only increase in high greenery was observed at points 5 and 7 for views "a" and points 1 and 2 for view "b". However, this is an increase of no more than 1%. It was also found that in 2019, a greater number of shrubs were visible inside the village than in 2009. In both analyzed directions at points 3 and 4, the proportion of medium height greenery was higher at that time in the southern direction by about 1% to 2% and in the northern direction by about 1% to 3%. In both sequences and both directions at point 4 the percentage of medium greenery was higher in 2019 than in 2009. Low greenery share inside the village (points 3, 4, and 5) remained at the same level, about between 2% to 6%, while at entries to the village its percentage share decreased in both the "a" and "b" view sequences.

It should be added that at the entries to the village of Psary, intensive development of large-size building structures of a service function started in 2000. Their percentage share in the analyzed views increased from 2009 to 2019, i.e., at point 1 "a" from 3 % to 12% (Figure 9). On the other hand, there were no industrial facility buildings in the northern part of the village in 2009, so point 7 did not include this then open space outside the built-up area of the village. Currently, these structures are out of the scope

of this point, therefore it is possible to establish another research point, no. 8, and to perform SALI analyses in 10 years' time in 2029.

The second qualitative part of the SALI (stage IIb) method is shown in Figures 10–12. It aimed to support the first part, especially in the case of subsequent meetings and work with stakeholders who can play a role in shaping the street landscape. Each analyzed sector in a given 15° view could receive a maximum of 12 points. A comparison of the assessment of views "a" and "b" at point 3 in 2009 and 2019 shows that the evaluation has changed to a lower one in places where the share of high greenery has decreased. Similar relationships were found in the evaluation of the remaining six sites located along Główna Street. The summary of the assessments of all the seven places evaluated (Figure 13) confirms that the greatest changes of assessment in the negative sense occurred in the outermost parts of the street. In its central part the evaluation remained similar.

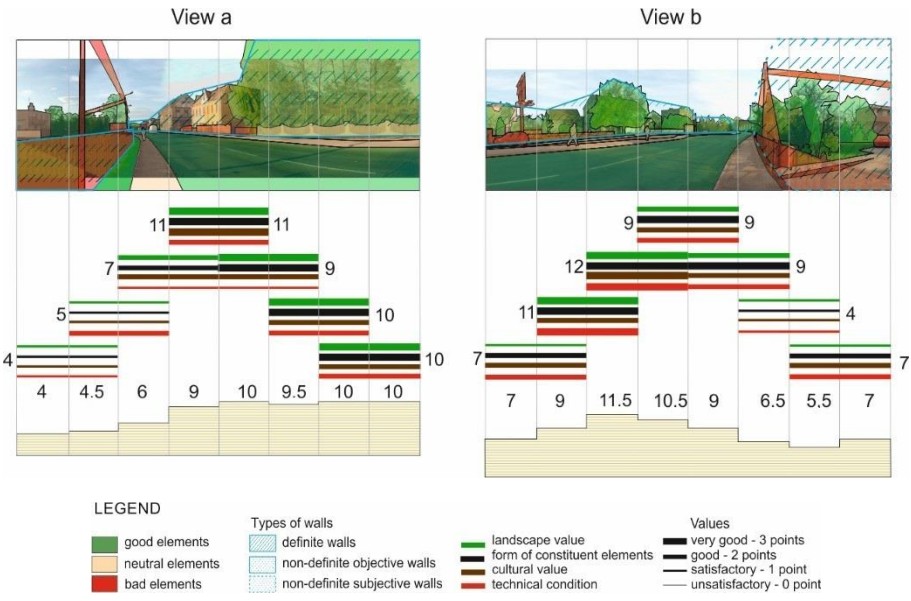

**Figure 10.** Stage IIb SALI qualitative analysis inside interior no. 3, view "a" and "b" in the autumn 2009 non-foliage period.

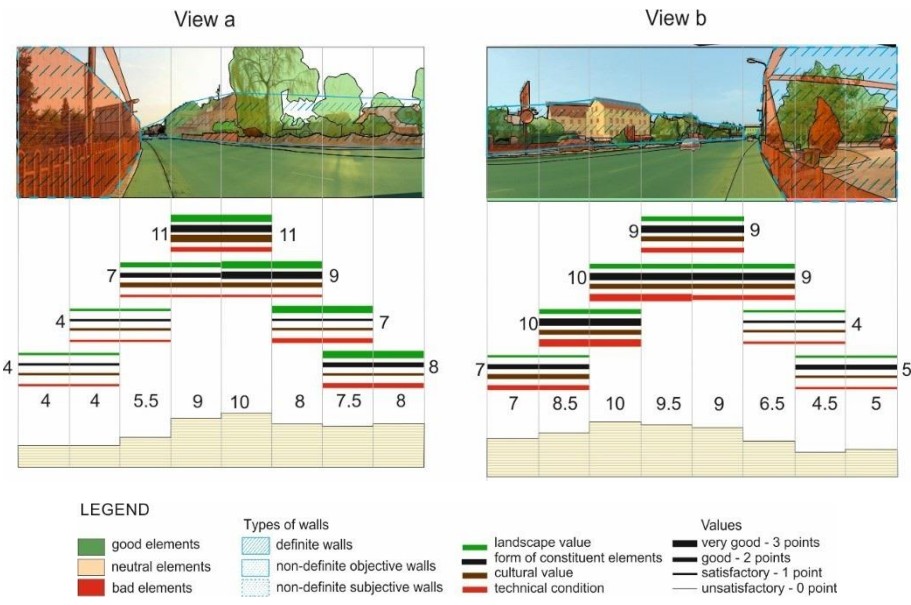

**Figure 11.** Stage IIb SALI qualitative analysis inside interior no. 3, view "a" and "b" in the autumn 2019 non-foliage period.

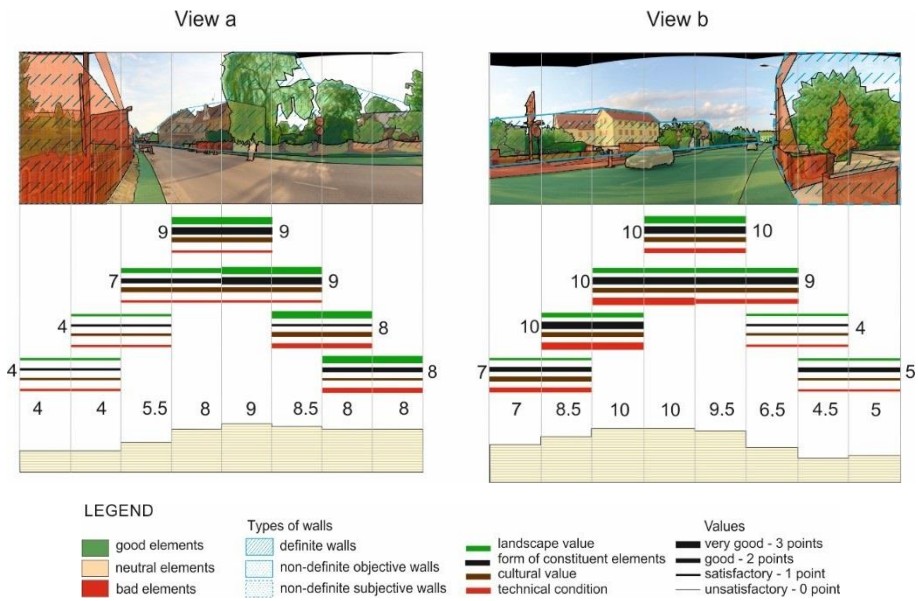

**Figure 12.** Additional qualitative analysis Stage IIb of the SALI for interior no. 3, view "a" and "b" in the summer 2019 foliage period.

The biggest change of assessment to a more negative one took place in interior no. 7, especially in view "a" (Attachment no. 3), and inside 1 and 2 in both views "a" and "b". Here, in addition to the reduction in greenery percentage share, an investment was made to change the road lane, i.e., sidewalk pavements were introduced and the roadway itself was widened. Additionally, large-size service and commercial facilities were built, which also affected the assessment.

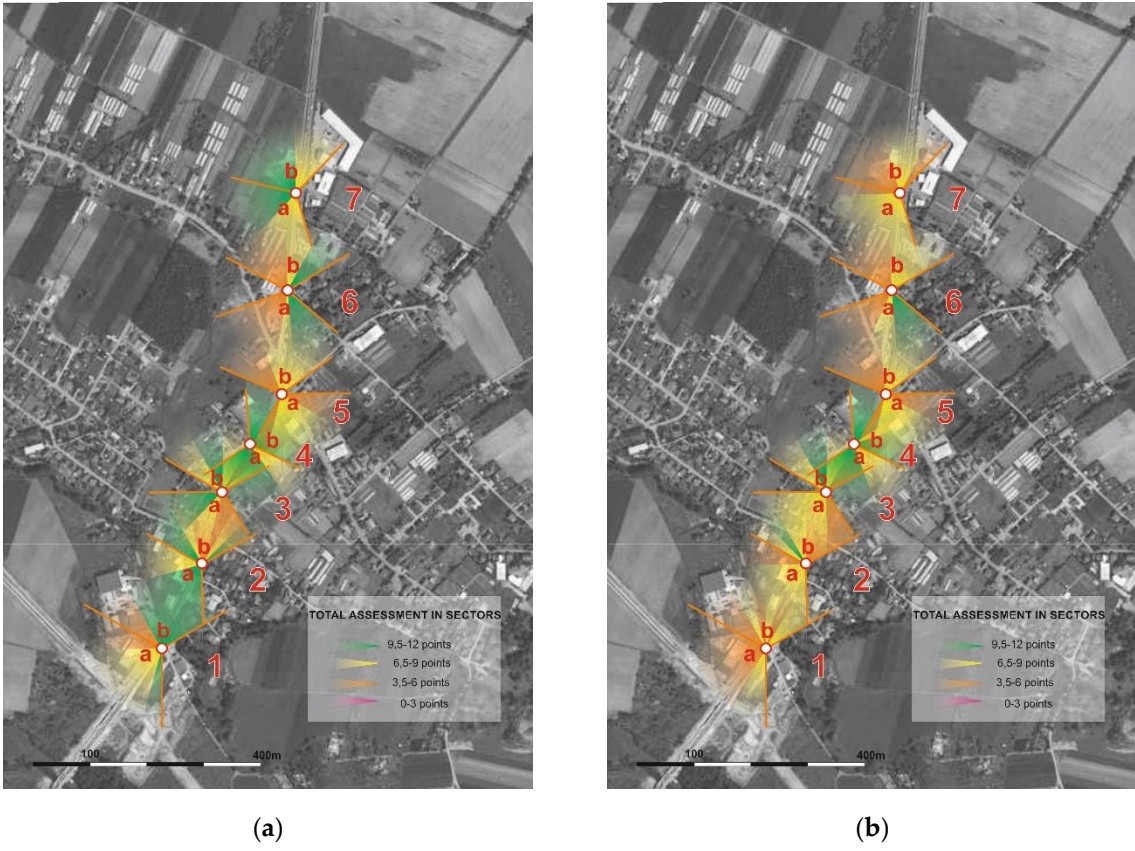

(**a**)                                                       (**b**)

**Figure 13.** Sectoral assessment shown on maps (SALI stage IIb)—(**a**) 2009; (**b**) 2019.

The summary of all seven sites on the evaluation maps (Figure 13) shows that the greatest changes in the negative aspect occurred in the outer parts of the street. The comparison with the graphs in Figures 8 and 9 showed that this was affected by significant losses of low greenery, and additionally at points 2, 3, and 6 by losses of high greenery. The summaries of all the analyses and results of the research conducted using the SALI method in seven points of Główna Street in the village of Psary in 2009 and 2019 are presented in Appendix A (Figures A1–A3). The changes in greenery management observed with the use of SALI and, above all, their visual clarity, will support the work of planners on the GI WFA, among others, in social consultations with local residents and other stakeholders.

## 4. Discussion

The term landscape interior applied in the SALI method used in Poland is close in meaning to the English term enclosure. The latter appeared in descriptions of composed historical landscapes [85], and entered the theory of landscape in the 20th century thanks to the works of Cullen [86] and Ashihara [87], who applied it to the city landscape. Later on, the concept of enclosure was adopted for the study and design of large-area compositions, e.g., agricultural-forest-settlement landscape systems [88,89], and is considered the basic spatial unit for the interpretation of landscape physiognomy [90].

In order to perform tests using the SALI method, it is necessary to take field photos. This makes it more labor-intensive than methods which are based on ready-made photos, e.g., GSV [34,39,41]. We must remember, however, the use of ready-made GSV images has its limitations due to the lack of information about the exact time of taking the photo. This is a disadvantage, especially in comparative studies of a given view. Also, the location of the horizon in the frame of such a photo cannot be controlled, because every attempt to move it beyond half of the frame causes distortion of the objects visible in the photos. Therefore, authors using GSV in their research, such as Li et al. [39] did not use their method to make comparisons of the same views at intervals, as proposed in this SALI method, but to analyze the views in different places at the same time. In the SALI method, however, it is possible to frame the photos in an arbitrary way. The most important is the repeatability of the method of framing in the compared views. This technique of quantitative and qualitative evaluation from the SALI method can be used, e.g., to compare old photos with printed contemporary ones. In the present study, the base material was photographs taken 10 years earlier, hence the contemporary ones had to be adjusted to them. What is important is that photographs in the SALI method can be taken during the non-foliation and foliation period, which additionally makes it possible to show the variability of the landscape over time, resulting from the passing of seasons.

The traditional counting method used in the SALI method for the area of landscape components may not be as precise, but is still sufficient to achieve the objectives. On the other hand, however, the development of computer techniques and the availability of various types of software may allow the quantitative calculation of the SALI method to be improved in the future. The current traditional way of performing calculations makes the method relatively labor-intensive and time-consuming.

Another issue in the human perspective studies is the scope of the visible and analyzed image: its width and depth. From maps and plans, as well as satellite images, it is primarily possible to obtain information on coverage forms and functions of individual landscape components. It is also possible to use viewshed analysis tools available in GIS. This makes it possible to determine the range of view from one or more points from a human perspective. However, it is difficult to define how individual landscape components affect the human perception of a landscape fragment. In most studies, all elements visible in the picture are taken into account regardless of their distance from the observer, e.g., an element at a distance of 1.0 m from the observer, or an element 100 m away from the observer. Only the confrontation of the view with the projection (map) allows the determination of the distance of the examined elements from the observer, and above all their accessibility. Research on the current state and changes in the landscape from a human perspective should complement and support the research carried out on maps and aerial photographs. Landscape elements and their changes

noticeable on the plan are not always synonymous with the visibility of elements and their changes from a human level. In the conducted at-eye-level research on the identification of landscape elements, including greenery, attention is also given to the accessibility related to visibility [28,41]. The presented SALI method can be enhanced by studies on accessibility to visible greenery. Thanks to mapping on the plan of the elements visible in the picture, it is possible to precisely determine in which area they are located—whether it is public, semi-public, or private. In this article this issue has not been developed, but it is possible to do so in the next stage of research. Knowledge gained through correlating the view with the plan would be valuable in formulating relevant provisions in local planning documents, in creating standards and guidelines for planning and designing greenery, buildings, infrastructure.

Until now, transparency of individual view components has not been taken into account in landscape research from a human eye-level perspective. For example, a fence, a tree or shrub crown, railway tracks, or power transmission lines do not constitute "full" elements (covering the entire view)—the elements behind them are to a greater or lesser extent visible. The transparency of objects in the foreground should be taken into account in quantitative analyses of at-eye-level, as it will determine the share of particular elements in a given view, but also the possibility of perceiving those elements which are seemingly covered up, but still visible in the non-foliage period, as shown in the study of Psary village using the SALI method.

The method presented in the paper does not take into account the volume of individual elements, but only their distorted perspective projection on a vertical or horizontal plane. However, it makes it possible to show the quantitative differences of individual elements in the image seen by a viewer, and also makes it possible to assess the impact of human activity on the landscape. This information can be used in the process of making land-use decisions, as an argument in discussions concerning, e.g., development projects of particular places, as well as on the form, size, or location of new buildings and increasing the share of greenery, especially high greenery in the analyzed landscape interiors.

The European Landscape Convention [8] places great emphasis on the assessment of the current state of the landscape and the changes occurring in it with the participation of the local communities. The issue of social participation in landscape research has been already addressed by many authors [91–96]. The presented SALI method can be used during workshops—their participants, under the guidance of an expert, can perform part of the work, e.g., take photos, create a matrix, make an initial quantitative and then qualitative assessment of individual elements. The way of landscape interior view analysis presented in the method does not require high quality equipment and can even be performed without computer programs, which can be considered an advantage when using this method with local community participation.

## 5. Conclusions

The results presented in the article confirmed the validity of using the SALI method to study landscape changes from a human perspective, with particular emphasis on the green element in the landscape. It is debatable whether the reduction in the number of trees found as a result of the analysis will be compensated with the local increase in the number of shrubs, especially in the context of continuity and communication with natural areas located outside the village. Lack of preventive measures or steps aimed at the introduction of well-thought-out greenery plantings will contribute to the weakening of the planned GI WFA system in the area around Wrocław and will deprive the inhabitants of Psary of the benefits resulting from the presence of greenery, especially high greenery (e.g., in terms of reduction of temperatures during heat waves, improvement of air quality, regulation of water circulation, and beauty of the landscape).

In the opinion of the authors, the conducted research as well as the obtained results prove that the SALI method can be used to monitor greenery changes in green infrastructure systems in urbanized areas. Its main value results from the possibility to show changes in greenery from the human level and thus complement the research on green infrastructure carried out with the use of maps, plans, or satellite images. The results of at-eye-level research on green infrastructure will be important for a

variety of stakeholder groups: both professionals—spatial planners and designers and representatives of public administration units, as well as local residents. For the former, professionally involved in spatial planning, the results of SALI analyses may be of assistance in making the right decisions on shaping greenery systems in urban areas, for the latter they can prove to be useful in discussions on making specific investments. As a landscape view recorded in the picture appeals more to human imagination than a picture on the map, it may be easier to understand, e.g., the need to introduce new plantings of high greenery in public areas and its significance in the landscape. The adequacy of the method for monitoring changes in green infrastructure systems is also justified by the fact that the method allows for conducting research in the landscape interiors of village streets. A street, on the other hand, is a space that crystallizes the plan of built-up areas of both cities and villages, and can therefore be used to build natural connections and support green infrastructure systems at various scales.

Monitoring landscape changes in the interiors of villages/small towns within the GI WFA is an important measure to support further management of this system. In the following years, it is advisable to undertake research in other similar villages located within the urban fringe of Wrocław, which will make it possible to determine the condition and changes of the landscape, including greenery (green infrastructure) in their areas. Summing up, it should be emphasized that the SALI method is a universal tool that can be widely used. It does not require any specialist equipment or data which might be difficult to obtain, e.g., under special permissions or competences. The authors see the application of the method used for research in other villages located in the urban fringe area not only around the city of Wrocław, but also in other large cities in the world. This will make it possible to monitor changes in the landscape and support its planning, with particular emphasis on greenery elements. In the future, the method shall be tested in the studies of the landscape of street interiors inside urban areas.

**Author Contributions:** Conceptualization, I.N.-F. and J.R.; methodology, I.N.-F., J.R and A.P.; formal analysis, I.N.-F, J.R.; investigation, I.N.-F., A.P., J.R., and J.P.; resources, I.N.-F..; data curation, A.P.; writing—original draft preparation, I.N.-F. and J.R.; writing—review and editing, I.N.-F. and J.R.; visualization, I.N.-F. and A.P. and J.P.; supervision, I.N.-F. All authors have read and agreed to the published version of the manuscript.

**Funding:** The research was financed from the funds for the statutory activities of the Institute of Landscape Architecture University of Wrocław Environmental and Life Sciences.

**Acknowledgments:** We would like to thank Paweł Filipiak who helped us in editing the drawings in ArcGIS.

**Conflicts of Interest:** The authors declare no conflict of interest.

## Appendix A

Figures A1–A3 present the summaries of all analyses and results of research conducted using the SALI method at seven points of Główna Street in the village of Psary in 2009 and 2019.

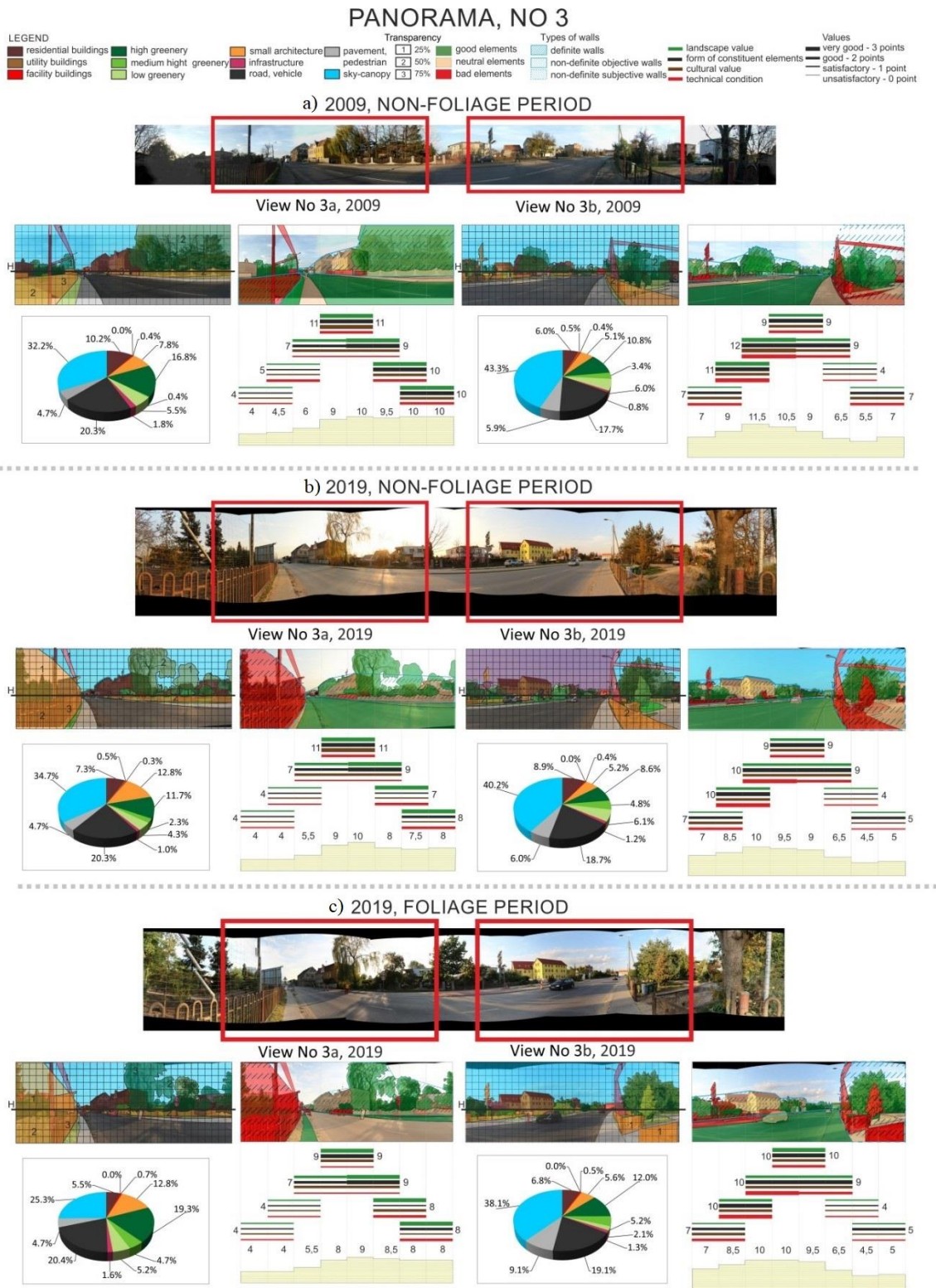

**Figure A1.** Summary of the results of SALI surveys for point no. 3 in the landscape interior of Główna Street in the village of Psary: (**a**) autumn 2009 non-foliage period; (**b**) autumn 2019 non-foliage period; (**c**) summer 2009 foliage period.

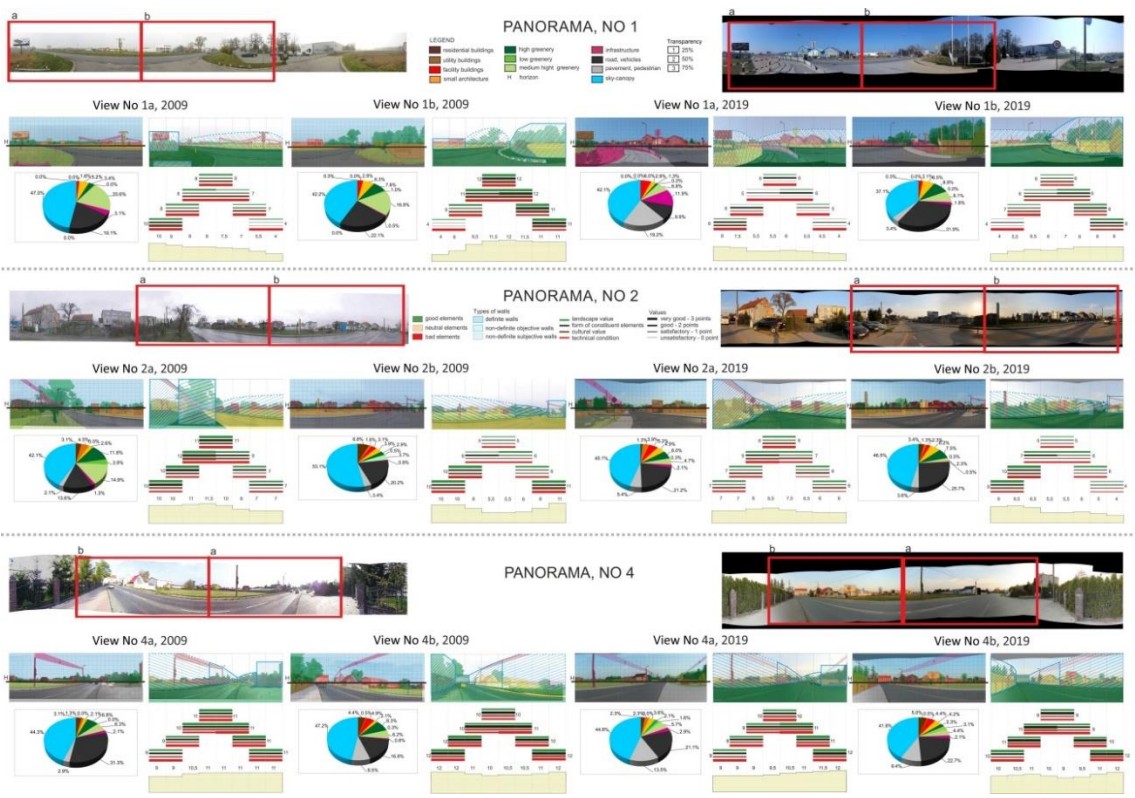

**Figure A2.** Total summary of the results of the SALI method for points 1, 2, and 4 in the landscape interior of Główna Street in the village of Psary in 2009 and 2019.

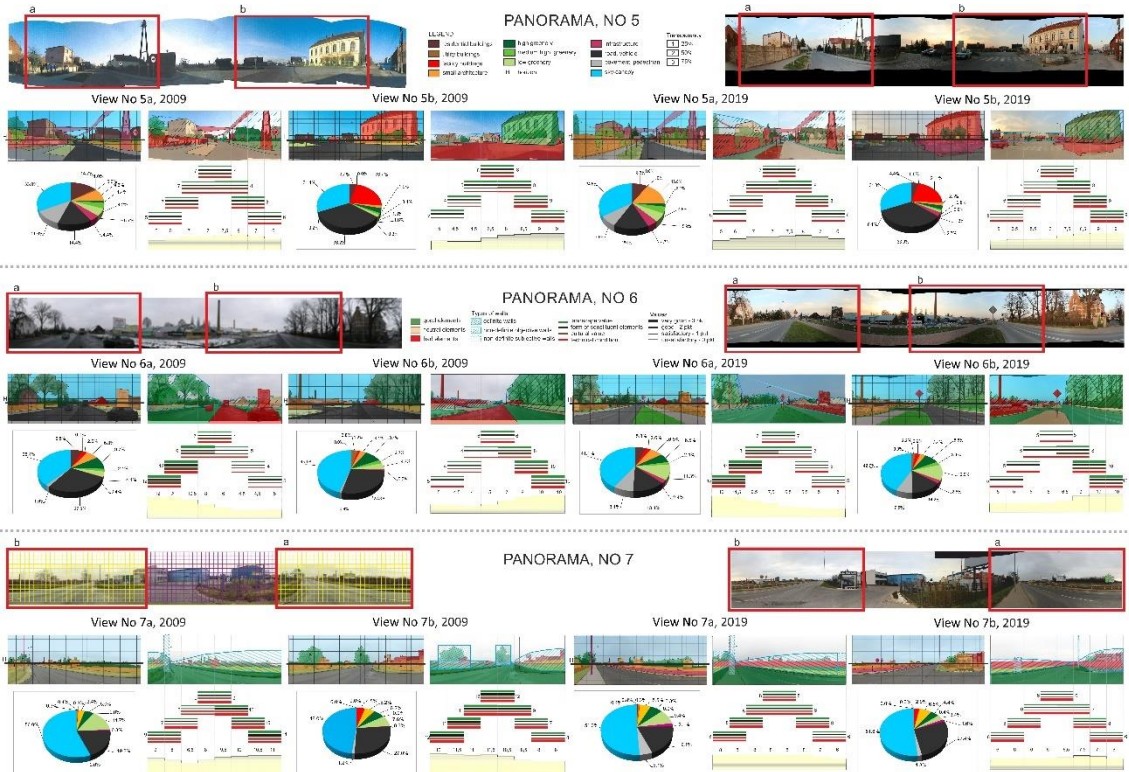

**Figure A3.** Summary of the results of the SALI method for points 5, 6, 7 in the landscape interior of Główna Street in the village of Psary in 2009 and 2019.

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
