# Peer review of "Sectoral Analysis of Landscape Interiors (SALI) as One of the Tools for Monitoring Changes in Green Infrastructure Systems"

_sustainability, doi:10.3390/su12083192_

Round 1

Reviewer 1 Report

The topic of the article is interesting and the results are really usefull. It it obvious that authors did a huge research in a long time period.

The only thing I would like to be changed is the Figure 2. Some parts are not easy to read or are covered and there is a mistake I think - Ddivision of views into 15 sectors.

I really appreciate the Discussion part.

Reviewer 2 Report

Generally, I appreciate this article as it refers to an important problem of landscape quality in small suburban towns, which is rather left out in the literature. The author(s) present a high-quality study.

Below please find the specific remarks:

  1. Introduction

41-41 Sounds better if you write, e.g., " Advances in remote sensing and geographic information systems...".

74-75 Please be careful when defining open areas as occurring at a regional scale. We can also talk about open areas on a city scale, for example.

In the introduction, please provide information concerning the SALI method you have used: Is it a new method? Who has developed it? What were the uses of the SALI until your research? Are there similar methods used anywhere in the world? Why did you choose the SALI method over another? Is SALI supposed to be better than other approaches in some aspect?

  1. Materials and Methods

2.1 Study Area

100-103 Figure 1 The figure title should not be part of the chapter. Here only information about what appears in the figure is given, without any comment.

116-118 A sentence starting with: "Unfortunately..." is very difficult to understand.

2.2. Method of Sectorial Analysis of Landscape Interiors (SALI)

167-168 The diagram shown in Figure 1 is only partially legible. Some editing errors are there (the text does not fit in the chart boxes, and the resolution in the reviewer's copy is insufficient.

In the method description, I recommend not to give colours that are used to identify individual objects. Specifying them in the Results chapter will naturally correspond to the figures presented there.

169-195 Each of the listed stages/steps should be introduced in the method diagram (Figure 2).

They can be inserted in the further text instead of being listed separately.

Please consider whether you can simplify the description of the methodology, e.g. rewrite verbose sentences.

  1. Results

374-378 The information that you have examined changes between the two terms needs to be given immediately at the beginning of the Methodology chapter.

380 Figure 3 The figure title should not be part of the chapter. Here only information about what appears in the figure is given, without any comment.

409-410 Figures 5-7 and Appendix 1. These illustrations are tough to read. I suggest that you increase their size (and perhaps resolution). Diagrams and legends are most difficult legible. Not all elements marked in the legends are visible on diagrams and views. You need to: place these objects on the views/diagrams, comment on their absence and/or remove them from legends. In Figure 5, the legend is vertically inverted.

422-423 Table 1 Consider zooming in this figure or indicating the specific values.

433-434 Table 2 The values displayed in the charts could be much more readable when removing the shadowing of the chart bars.

  1. Discussion

537-539 Viewshed analysis tools are available in GIS programs to check the range of view from the perspective of one or more observers. This is done using the Digital Surface Model (DSM), most often made from LIDAR data.

557-567 Please specify (in the Results chapter) the way you used the transparency of the view components.

  1. Conclusion

582-585 Did I understand correctly that: based on the analysis of one street in the Psary village, you concluded that no actions were taken to manage the Green Infrastructure of the Wroclaw Functional Area in 10 years?

At the end of this chapter, it is worth adding what the outlook for your research is.

Reviewer 3 Report

This is an interesting study that has developed a new method for analysing photographs to provide a quantitative and qualitative analysis of landscapes, over a 10 year period in Poland. The results are interesting, a provide a unique way of quantifying this change at the human perspective. This study could make a strong contribution, and I see this concept being employed widely at the fine-scale, like the authors state. However, in it’s current form, this article is long and wordy, and is caught between it’s two aims 1) describing the new method 2) applying it as a case study. The authors switch back and forth between the two, which makes it hard to follow. Until the methods, it was not evident the authors were describing a new method. This is where the novel aspect of your research is, and I suggest focusing in on this as much as possible. Subsequently, the article needs a thorough rewriting of all sections. Detailed comments on the main components below, followed by minor ones.

Introduction – At no point do you introduce SALI. You posit the research gap for studying greenness, but you don’t introduce the method, nor introduce alternative methods. This needs to be rectified, as omitting a literature review on research that has explored this is a limitation of the article in it’s current format. You do go into a lot of this material in your discussion. In fact the first two paragraphs of your discussion could easily fit within your literature review, as they address the main ways this type of research has been undertaken.

The methods section is long and verbose. Firstly, section 2.1 is too long in my opinion. The long section on study area gives this paper a very local feel, which could limit the readership of such a paper. Some of this could be moved to the results/discussion section to support your results. Similarly the outline of the methods is too wordy. You’re writing in long sentences, with multiple commas, starting and stopping. There is a lot of reference to unexplained terms (e.g., golden ratio) and the long detail about how features are recorded makes it difficult to follow. I think a categorisation into landscape features could be condensed into a table (you start to do this in stage II through bullet points), and you could elaborate exactly how you classified each object, which would make it easier for readers to reference. Figure 1 is a good start, but you could go a lot further with this to identify the exact methods within phase II, particularly given the lengthy text description.

In the first part of the results, you re-reference the methods, talking through the steps of what you did. This isn’t needed and makes the article unnecessarily even longer. This is where you should be referencing your local case study, and suggesting the landscape reasons why you observed the results. As I’ve stated already, much of the discussion is literature review material, so can be shortened a lot.

Minor comments

GI as an acronym is much more readily used for ‘Geographic Information’. Given you discuss GIS within your article, I suggest spelling out all acronyms of GI, as it’s confusing and potentially masks the main message of your paper.

L19 – Shown on – change the verb to illustrated, applied, etc

L20 – Give the country of research. Most readership won’t necessarily know where Psary is

L22 – According to the authors is awkward. Change to something along the lines of “The results of the study indicate”

Abstract is largely missing results or discussion items, and really only introduces the methods. This needs to be rewritten to follow abstract formats.

L28 – Landscape should be either “A landscape” or “Landscapes are”

L35 – too many uses of ‘context’

L42 – spell out GIS the first time you use the acronym

L49 – ‘in terms of sense of security’ is awkwardly posed

L58-61 – implies that the researchers will complement results with GIS. If you don’t use this as discussion material

L88 – ditto comment on the country

L90-91 – you don’t need to keep spelling out the acronym

Figure 2 – the choice of colours for the boxes is confusing. Does the Village box have the same use as your stages? That is the implication of the colour scheme.

L203-205 – Is this part of the methods used in this study?

Table 1 is a figure – I think the numbers are overkill. They’re too small to read, and most people will get the general idea of what pie charts are showing.

Table 2 is also a figure. Values are too small to read, but it shows an interesting change.

I haven’t gone through the results and discussion with a fine comb given the rewrite needed.

Reviewer 4 Report

Landscape is an important component of human space, and the changes taking place in it condition our well-being on the one hand, and on the other, affect the functioning of the entire natural environment system.

The search for new ways to monitor changes in the landscape is extremely important, especially in areas of suburbanization. The spatial structure of these areas has a direct impact on the processes occurring in the cities with which they are directly adjacent

The paper presents one of the methods of analysis and assessment of changes occurring in suburban landscapes on the example of the village of Psary, located near WrocĹ‚aw (Poland), which, in the authors' opinion, can be used to monitor landscape changes in green infrastructure systems.

Given the wide range of metods developed for this purpose in previous research, such a study is relevant and it has some important
practical implications. However, there are some minor issues which, in my opinion, require revision.

Please find my comments and suggestions to each section of the manuscript below.

Abstract
The background, objectives of the study, methods and key findings are clearly outlined.

  1. Introduction
    The introduction is quite general but it provides a proper background for the study and a sufficient review of the current literature, although some issues requires further clarification (see detailed comments).

Detailed comments

  • line 78-81 - please provide relevant references
  • line 77 - please expand the shortcut

2. Materials and Methods

I find the methods used for the study and its design generally justified and well described. The structure of this section is clear and sufficient references are provided.
Detailed comments:

The description of the research area lacks its general location (not everyone knows where Wrocław is located)

Figure 1.

The figure should be corrected. The left part is not understood. The location of the city of Wrocław should be marked there and the legend should be placed on the map. It would be worth placing a map of the country or continent with the location of the research area marked.

In turn, the part on the right does not even have a reference scale

  • line 135 - why 2.2? - check the numbering of the subsections

Figure 2.

The construction of the diagram is correct, but the graphic representation should be improved. The "taking 2 views a and b" step covers other rectangles.

  • line 169, 176 - check the numbering of the subsections

Results

Figure 3 and 11

no reference scale

The results lack a reference to the green infrastructure of the city of Wrocław. How does monitoring the panoramas of the village of Psary affect the activities aimed at preserving this infrastructure? There is a lack of a graphical representation of which fragment of the Wrocław GI is the studied area.

Round 2

Reviewer 3 Report

The article reads much clearer and succinct in places. The position of the paper is no longer ambiguous and it’s clear where the research gap of the article is. Well done. Make sure you proof read all the new material. Some minor examples of grammar below.

L61-64 - can be clarified

L127-128 - page number needed for direct quotes

L177 - missing full stop

L454 - sentence reads awkwardly